# Downsizing and purchases of psychotropic drugs: A longitudinal study of stayers, changers and unemployed

**Sandra Blomqvist** [1]*, **Kristina Alexanderson**[2], **Jussi Vahtera**[3], **Hugo Westerlund**[1,2], **Linda L. Magnusson Hanson**[1]

1 Stress Research Institute, Stockholm University, Stockholm, Sweden, 2 Department of Clinical Neuroscience, Karolinska Institutet, Stockholm, Sweden, 3 Department of Public Health, University of Turku, Turku, Finland

* sandra.blomqvist@su.se

## Abstract

### Background

The evidence is insufficient regarding the association between organizational downsizing and employee mental health. Our aim was to analyze trajectories of prescribed sedatives and anxiolytics with a sufficiently long follow-up time to capture anticipation, implementation and adaption to a downsizing event among stayers, changers and those who become unemployed compared to unexposed employees.

### Method

Residents in Sweden aged 20–54 years in 2007, with stable employment between 2004 and 2007, were followed between 2005 and 2013 (n = 2,305,795). Employment at a workplace with staff reductions ≥18% between two subsequent years in 2007–2011 (n = 915,461) indicated exposure to, and timing of, downsizing. The unexposed (n = 1,390,334) were randomized into four corresponding sub-cohorts. With generalized estimating equations, we calculated the odds ratios (OR) of purchasing prescribed anxiolytics or sedatives within nine 12-month periods, from four years before to four years after downsizing. In order to investigate whether the groups changed their probability of purchases over time, odds ratios (OR) and their 95% confidence intervals (95% CI) were calculated contrasting the prevalence of purchases during the first and the last 12-month period within four time periods for each exposure group.

### Results

The odds of purchasing psychotropic drugs increased more for changers (sedatives OR 1.08, 95% CI 1.05–1.11) and unemployed (anxiolytics OR 1.08, 95% CI 1.03–1.14), compared to unexposed before downsizing, while for stayers purchases increased more than for unexposed during and after downsizing. Among those without previous sickness absence, stayers increased their purchases of psychotropic drugs from the year before the event up to four years after the event.

**Data Availability Statement:** Data cannot be shared publicly because of Swedish Ethical Review Act, the Personal Data Act, and the Administrative Procedure Act as well as General Data Protection

Regulation GDPR because of the sensitivity of the data. The micro level data on employment, workplace downsizings and demographic characteristics utilized in our study are available from the Statistics Sweden (mikrodata@scb.se) for researchers who meet the criteria for access to confidential data. Further information about author criteria and instructions on how to apply for data can be found at https://www.scb.se/en/services/ordering-data-and-statistics/ordering-microdata/ Correspondingly, individual level data on psychotropic drug purchases is available through the National Board of Health and Welfare in Sweden and researchers can apply for data using the following email adress: Registerservice@socialstyrelsen.se. Further information about requirements and instructions on how to apply can be found at: https://www.socialstyrelsen.se/en/statistics-and-data/statistics/.

**Funding:** The study was funded by the Swedish Research Council for Health, Working Life and Welfare recieved by the Stockholm Stress Center [grant number 2009-1758]. http://forte.se/en/ The funders had no role in study design, data collection and analysis, decision to publish, or preparation of the manuscript.

**Competing interests:** The authors have declared that no competing interests exist.

## Conclusion

This study indicates that being exposed to downsizing is associated with increased use of sedatives and anxiolytics, before the event among those who leave, but especially thereafter for employees who stay in the organization.

## Introduction

Downsizings has become a common aspect of working life today, but may be associated with negative health consequences [1]. According to previous research, the negative health consequences of downsizing may be attributed to job insecurity [2, 3], and in the case of job loss, loss of income, social support contact with colleagues and/or social status could be the underlying factors of poor health [4, 5]. For those remaining within the organizations increased workload after the downsizing event [6, 7] and guilt over the dismissal of co-workers, are often put forward as explanations [8].

So far, most studies on downsizing and mental health have used self-reported measures on either exposure, outcome or both [2]. To the best of our knowledge only two studies have used more objective measures of downsizing and mental health comparing survivors and those who lost or left their job to unexposed workers [9, 10]. A Finish study on municipal employees [9] found that men who lost/left their jobs and men and women who stayed in an organization after downsizing had higher rates of purchasing psychotropic drugs compared to the unexposed. In our own previous work, we found that the odds of purchasing prescribed antidepressants increased more during and after the downsizing for exposed workers without prior health problems than for unexposed workers [10]. People with prior health problems and unemployment following downsizing increased their purchases of antidepressants before the event. This indicates that it is imperative to study people's health across a relatively long time window and account for possible health selection. Still, the evidence is insufficient regarding the association between organizational downsizing and employee mental health due to cross sectional designs and the large heterogeneity of measures, sample and designs in previous studies [11]. Findings have been inconclusive, especially for survivors [5, 9, 12]. In the previous research there has also been a strong focus on depression while less is known about how downsizing affect other mental health problems such as anxiety or sleep disturbances.

Within this prospective register-based cohort study we aimed to analyze trajectories of prescribed sedatives and anxiolytics for employees exposed to downsizing, i.e. stayers, changers and those who become unemployed after downsizing, compared to unexposed employees. Because anxiety and sleeping problems have been found to precede the onset of depression [13–16], we hypothesized that downsizing may increase purchases of anxiolytics and sedatives in an earlier stage than in purchases of antidepressants. We first aimed to examine whether downsizing increases purchases of sedatives and anxiolytic drugs and second, given that they already purchased these types of drugs during the study period we aimed to examine whether downsizing was associated with increased numbers of purchases of these drugs, which may indicate an increased severity of symptoms. The present study is the correct version of a previously published version of this paper, now retracted, in which we found errors due to a coding mistake [17, 18]. That version had erroneously classified some employees as stayers, when they in fact had changed employer the year following downsizing. The error arose due to a mistake of relying on workplace, rather than individual, level information when classifying the employees' post-downsizing status. The corrected version therefore has a larger proportion of

changers, a smaller proportion of stayers, while the unemployed remain unaffected by our error. Consequently, the results for the changers became more stable after the correction. Syntaxes and results have been carefully checked, cross-checked with proportions in public administrative data, and validated by studying expected changes in a randomly chosen, large company.

## Materials and methods

### Study population

The study population was drawn from the Longitudinal integration database for health insurance and labor market studies (LISA), administered by Statistics Sweden, including all people aged 16 to 64 years and living in Sweden on December 31, 2004 (n = 5,750,279). Our analytical sample contained those who lived in Sweden between 31 December 2004 and 31 December 2013, were gainfully employed (i.e., with income from work above a specific amount) each year between 2004 and 2007, employed according to their primary occupation the year before they were exposed to downsizing and aged 20 to 54 years in 2007, i.e., those who had entered the labor market and were not about to retire. Information from the Swedish National Prescribed Drug Register and Statistics on Dynamics of Enterprises and Establishments (DEE) was obtained by linkage through the Swedish personal identity number but de-identified by Statistics Sweden before delivery of data and used to determine exposure to downsizing. People working in an establishment that was not registered in DEE, were excluded. Those who had been exposed to a downsizing event between November 2006 and November 2007, the year before our period of interest, were excluded to enable us to study new downsizing events. This resulted in a total analytic sample of 2,305,795 individuals, for details see Fig 1. The

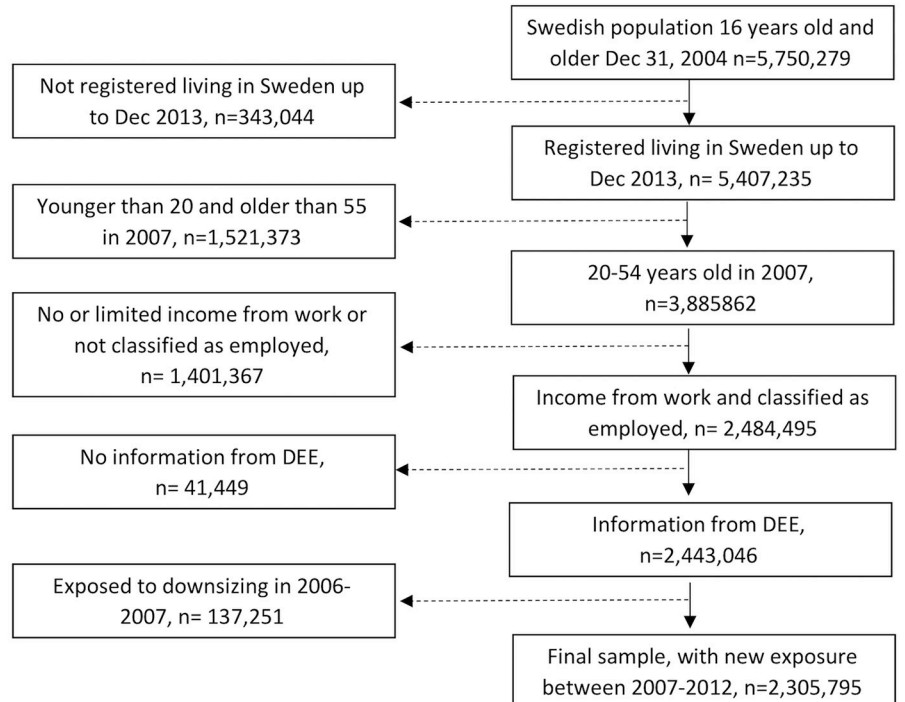

**Fig 1. Flow chart of study sample, based on information from the Longitudinal integration database for health insurance and labor market studies (LISA) and Statistics on Dynamics of Enterprises and Establishments (DEE).**

project has received ethical approval from the Regional Ethical Review Board of Stockholm and the data were analyzed anonymously.

## Exposure

The Statistics on Dynamics of Enterprises and Establishments (DEE) register provides information on structural changes within establishments and organizations. Through information from the LISA register on main employer, collected annually on 30 November, it was possible to link persons to the DEE register and to assess if a person had been employed at a workplace that went through a major downsizing. In accordance with the most commonly used definition, we classified employees as exposed to major downsizing if they worked at an establishment that reduced their staff by ≥18% between November 2007 and Nov 2008, Nov 2008 and Nov 2009, Nov 2009 and Nov 2010, or Nov 2010 and Nov 2011, respectively [1, 5, 10]. Only new events were examined, i.e., exposure in one 12-month period without exposure in the previous one. Those unexposed to downsizing (n = 1 390 334) were randomly assigned to a cohort with a 12-month period used as unexposed reference and follow-up matching the periods above, to get a suitable comparison group. Employees in the unexposed group had equal probability of being assigned one of the four sub-cohorts.

Those exposed to downsizing were grouped into three categories, depending on their employment situation after the downsizing. Those, who have been referred to as survivors in previous literature were divided into two groups; 1) "stayers", consisting of those who stayed in the same organization the year after the downsizing and 2) "changers", containing those who remained employed but changed organization after the downsizing. Those who at the end of the year of downsizing were not gainfully employed or had received more than 180 days with unemployment benefits were categorized as "unemployed".

## Outcome

In this study, purchases of prescribed anxiolytics and sedatives were used as outcomes. Information on drugs was obtained from the Swedish National Prescribed Drug Register covering November 2005 to November 2013. All prescriptions coded N05B (anxiolytics) and N05C (hypnotics and sedatives) according to the Anatomical Therapeutic Chemical classification were extracted from the register with information about date the prescription was filled. This information was used to identify all 12-months periods that a person had purchased prescribed anxiolytics (yes/no) or sedatives (yes/no) during a 9-year time window. We centered the data around the 12-month period a person was exposed to downsizing (year 0) and used available data on annual purchases of prescribed sedatives and anxiolytics from four years before to four years after a downsizing event. These variables were used to investigate if exposure to downsizing increased the likelihood of purchasing sedatives and anxiolytics. The annual numbers of purchases of anxiolytics and sedatives were studied among the users, i.e., those who at any point during our study period had bought anxiolytics or sedatives, to assess the association between downsizing and the amount of purchases.

## Covariates

Annual information on sociodemographic factors that are plausibly associated with downsizing and mental health was retrieved from the LISA register between 2004 and 2013. We controlled for sex, age, educational level, type of living region, and family situation. Men and women tend to work in different sectors that might be associated with different working conditions [19] and gender differences have been observed for depressive symptoms [20]. Furthermore, worrying about losing your job seem to have more detrimental consequences among

older employees compared to younger employees [21], and purchases of psychotropic drugs increase with age [22]. Age was categorized into 20–34, 35–49, and 50–59 years, in order to further test for interaction. Low socioeconomic position is associated with higher prevalence of common mental disorders [23] and unemployment is more common among low educated [24]. Educational level was therefore used and categorized into groups: (1) Primary education, (2) Secondary education, (3) Post-secondary education <3 years, and (4) Post-secondary education ≥3 years. Regional differences have been observed in mental health [25] and the consequences of unemployment might be somewhat different in rural areas where new job opportunities often are more limited compared to in bigger cities. Type of living region was coded into three categories depending on the size and density of the population in the municipality where the person lived: (1) Big cities (metropolitan areas, with >90 000 inhabitants within a 30 km radius from the largest municipality center), (2) medium-sized cities (areas with 27 000 to 90 000 inhabitants within a 30 km radius from the largest municipality center, and (3) small cities/villages (areas with <27 000 inhabitants within a 30 km radius from the largest municipality center). Social support has been regarded as a protective factor for both depression and negative health consequences of unemployment [26], while having children may introduce a work to family conflict affecting mental health negatively [27]. Family situation was divided into four different categories: (1) Living with a partner (married, registered partner, or cohabiting) and children, (2) Living with a partner but without children at home, (3) Living with children only and (4) Living alone. We also obtained information about the participants' pre-downsizing sickness absence history. In Sweden, an employee is generally entitled to sickness absence benefit from the Social Insurance Agency after 14 days with sick pay paid by the employer, if unable to work due to morbidity. We defined previous sickness absence as one day with sickness absence benefit or disability pension during the two years before a downsizing event (meaning at least 15 days of sickness absence).

## Data analysis

To examine the annual prevalence of people with purchases of prescribed sedatives and anxiolytics in the 9-year time window, we applied repeated-measures logistic regression using generalized estimating equations (GEE), which takes into account the intra-individual correlation between measurements. The trajectories were based on 9 observations per person and missing information was handled through full information maximum likelihood. In order to determine which correlation structure described the intra-individual correlation the best, Quasi-likelihood under the independence model criterion from analyses with different correlation structures were compared [28]. The autoregressive correlation structure showed the lowest value and was thus chosen. In order to study the changes in the amount of annual purchases of prescribed sedatives and anxiolytics, we applied repeated-measures Poisson regression analysis with generalized estimating equations estimation. In all analyses, trajectories were derived separately for the four groups.

After examination of the time trends, the 9-year time window was divided into four distinct periods: "early pre-downsizing" (years -4 and -3), "late pre-downsizing" (-3 to -1), "peri-downsizing" (-1 to +1) and "post-downsizing" (+1 to +4). In order to investigate whether the groups changed their probabilities of purchases over time, odds ratios (OR) and their 95% confidence intervals (95% CI) were calculated contrasting prevalence of purchases during the first and the last 12-month period in each of the four time periods for each exposure group. Crude analyses, only adjusted for calendar time, were followed by analyses controlling for sociodemographic factors and previous sickness absence, added to the model as main effects. We further tested for differences in time trends between the exposed groups and the unexposed group within the four periods. See S1 Appendix for the SAS 9.4 scripts used for these analyses.

We stratified the results by sex and age groups, respectively, to test for differences between men and women and across the age groups. Finally, in order to account for a health selection effect we stratified our sample by pre-downsizing sickness absence and conducted the analyses separately for employees with and without a previous sickness absence.

## Results

Within this population-based Swedish cohort study of 2,305,795 employees, 915,461 individuals (40%) were defined as exposed to downsizing. The majority of them were still employed at the same workplace the year after the event (Table 1). Among leavers, 22% became unemployed while the rest changed their workplace. Compared to the other groups, the unemployed were generally younger, lower educated, more often living alone and had more often had previous sickness absence.

As shown in Table 1, a larger proportion of the changers belonged to the youngest age group, but in general purchases of psychotropic drugs and socio-demographic factors did not differ much between unexposed, stayers and changers. About 12% of them bought any prescribed anxiolytics during the studied period and the average person made less than 1 purchase across the nine 12-month periods. Among the unemployed however, 20% bought anxiolytics and the average person made almost 2 purchases. Purchases of sedatives were slightly more common than anxiolytics in all groups, but the same pattern was found when comparing prevalence across different groups.

**Table 1. Descriptive statistics by employment status following from downsizing.** Sociodemographic factors from the year before the downsizing event and proportion of people with purchases of sedatives and anxiolytics across the studied period and average number of purchases made.

| Employment status after major downsizing | Unemployed | | Stayer | | Changer | | Not exposed | |
|---|---|---|---|---|---|---|---|---|
| | n | %(sd) | n | %(sd) | n | %(sd) | n | %(sd) |
| | 82,518 | | 544,031 | | 288,912 | | 1,390,334 | |
| Purchased any sedatives | | 20.3 | | 12.9 | | 13.9 | | 14.2 |
| Purchased any anxiolytics | | 19.6 | | 11.9 | | 12.7 | | 12.5 |
| Average number of purchases for sedatives | 2.8 | (14.8) | 1.2 | (8.1) | 1.2 | (8.2) | 1.4 | (9.3) |
| Average number of purchases for anxiolytic | 1.8 | (11.0) | 0.7 | (5.6) | 0.7 | (5.9) | 0.8 | (6.4) |
| Men | 42,870 | 52.0 | 292,550 | 53.8 | 153,583 | 53.2 | 715,967 | 51.5 |
| Women | 39,648 | 48.1 | 251,481 | 46.2 | 135,329 | 46.8 | 674,367 | 48.5 |
| 20–34 | 27,127 | 32.9 | 128,779 | 23.7 | 89,253 | 30.9 | 290,412 | 20.9 |
| 35–49 | 38,151 | 46.2 | 293,035 | 53.9 | 147,738 | 51.1 | 752,886 | 54.2 |
| 50–59 | 17,239 | 20.9 | 122,217 | 22.5 | 519,21 | 18.0 | 347,036 | 25.0 |
| Primary education | 13,501 | 16.4 | 57,875 | 10.7 | 24,682 | 8.6 | 124,000 | 8.9 |
| Secondary education | 51,899 | 63.1 | 307,766 | 56.7 | 150,322 | 52.1 | 675,064 | 48.6 |
| Post-secondary education <3 years | 5,134 | 6.2 | 36,599 | 6.7 | 20,913 | 7.3 | 96,678 | 7.0 |
| Post-secondary education ≥ 3 years | 11,759 | 14.3 | 141,033 | 26.0 | 92,677 | 32.1 | 492,926 | 35.5 |
| Living with partner. children at home | 6,484 | 7.9 | 48,480 | 8.9 | 21,541 | 7.5 | 131,237 | 9.4 |
| Living with partner. no children at home | 34,753 | 42.1 | 297,554 | 54.7 | 153,133 | 53.0 | 770,701 | 55.4 |
| Living only with children at home | 9,801 | 11.9 | 49,500 | 9.1 | 26,794 | 9.3 | 124,605 | 9.0 |
| Living alone | 31,480 | 38.2 | 148,497 | 27.3 | 87,444 | 30.3 | 363,789 | 26.2 |
| Living in big city | 24,035 | 29.1 | 187,997 | 34.6 | 119,888 | 41.5 | 535,327 | 38.5 |
| Living in medium-sized city | 30,638 | 37.1 | 193,330 | 35.5 | 96,330 | 33.3 | 492,457 | 35.4 |
| Living small city/village | 27,845 | 33.7 | 162,704 | 29.9 | 72,694 | 25.2 | 362,550 | 26.1 |
| Sick leave previous 2 years | 24,419 | 29.6 | 86,484 | 15.9 | 48,246 | 16.7 | 228,875 | 16.5 |
| Disability pension previous 2 years | 6,033 | 7.3 | 8,918 | 1.6 | 3,676 | 1.3 | 31,491 | 2.3 |

## Purchases of prescribed psychotropic drugs in relation to a downsizing event

Fig 2 presents trajectories of the estimated prevalence of purchasing prescribed anxiolytics in the 9-year time window, adjusted for calendar year, socio-demographic factors and previous sickness absence. Stayers, changers and non-exposed had a lower prevalence and a relatively stable pattern compared to the unemployed. However, when contrasting the odds of purchases of prescribed anxiolytics over time within groups, stayers appeared to have an increasing prevalence in the late pre- (OR 1.03 95% CI 1.01, 1.06), peri- (OR 1.05 95% CI 1.03, 1.08) and post-downsizing phase (OR 1.04, 95% CI 1.00, 1.07), see Table 2. The trend was also more pronounced than for unexposed in the peri- and post-downsizing phase. The * sign in Table 2 indicates when test result displayed significant trend difference between exposed and unexposed employees. Although the unemployed had a higher prevalence of purchasing prescribed anxiolytics across the whole period, they had a declining trend. However, the decrease seemed to be interrupted by the downsizing event as the prevalence increased with about 8% (OR 1.08, 95% CI 1.03, 1.14) during the late pre-downsizing period and then continued to decrease again in the peri-downsizing period (OR 0.84, 95% CI 0.80, 0.88) and the post-downsizing period (OR 0.91, 95% CI 0.85, 0.97), all of which were different from the pattern among unexposed. No clear trend was evident for the changers. The same analyses were performed for sedatives, showing relatively similar patterns. However, changers seemed to have larger increases in prevalence during the late pre- and post-downsizing phases compared to the unexposed, but less pronounced during peri-downsizing (Table 2 and Fig 3).

The average number of purchases of anxiolytics among users, S1 Fig, was higher for unemployed compared to the unexposed during the late pre-downsizing period, when adjusted for

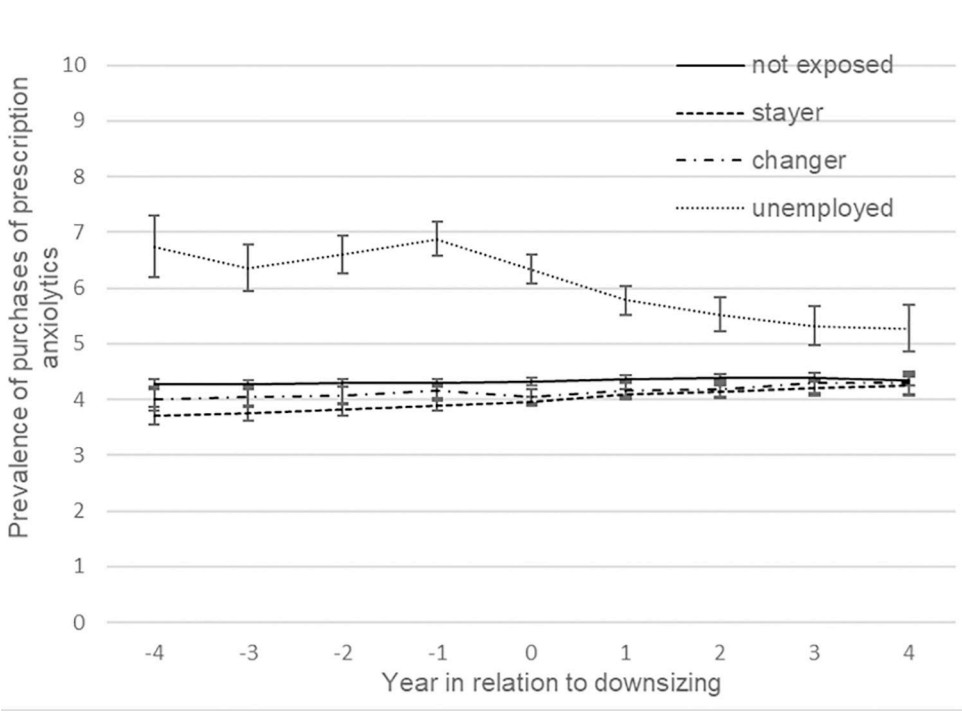

**Fig 2. Prevalence of purchased prescribed anxiolytics by employment status (%).** Adjusted for sociodemographic factors and previous sickness absence, with 95% CI.

**Table 2. Odds ratios (OR) for purchases of prescription anxiolytics and sedatives and their 95% confidence interval (CI) in relation to a downsizing event, comparing changes over time within groups in analyses stratified by downsizing and employment status after downsizing.** Any statistically significant differences between exposed and not exposed are expressed with * in mutually adjusted models.

| | Early pre-downsizing | | Late pre-downsizing | | Peri-downsizing | | Post-downsizing | |
|---|---|---|---|---|---|---|---|---|
| | '-3 to -4'[d] | | '-1 to -3'[e] | | '+1 to -1'[f] | | '+4 to +1'[g] | |
| | OR | 95% CI | OR | 95% CI | OR | 95% CI | OR | 95% CI |
| **Anxiolytics** | | | | | | | | |
| *Crude*[b]: | | | | | | | | |
| *Not exposed* | 1.00 | 0.98, 1.01 | 1.00 | 0.99, 1.02 | 1.02 | 1.00, 1.03 | 0.99 | 0.98, 1.01 |
| *Unemployed* | 0.92 | 0.87, 0.97 | 1.04 | 0.99, 1.10 | 0.81 | 0.77, 0.85 | 0.86 | 0.81, 0.92 |
| *Stayer* | 1.00 | 0.97, 1.02 | 1.03 | 1.00, 1.06 | 1.04 | 1.02, 1.07 | 1.02 | 0.98, 1.05 |
| *Changer* | 1.00 | 0.96, 1.04 | 1.02 | 0.98, 1.05 | 0.99 | 0.96, 1.03 | 1.01 | 0.97, 1.05 |
| *Adjusted*[c]: | | | | | | | | |
| *Not exposed* | 1.00 | 0.98, 1.01 | 1.01 | 0.99, 1.02 | 1.02 | 1.00, 1.03 | 0.99 | 0.98, 1.01 |
| *Unemployed* | 0.94* | 0.90, 1.00 | 1.08* | 1.03, 1.14 | 0.84* | 0.80, 0.88 | 0.91* | 0.85, 0.97 |
| *Stayer* | 1.01 | 0.98, 1.04 | 1.03 | 1.01, 1.06 | 1.05* | 1.03, 1.08 | 1.04* | 1.00, 1.07 |
| *Changer* | 1.01 | 0.97, 1.05 | 1.03 | 0.99, 1.07 | 1.00 | 0.97, 1.04 | 1.03 | 0.99, 1.07 |
| **Sedatives** | | | | | | | | |
| *Crude*[b]: | | | | | | | | |
| *Not exposed* | 0.99 | 0.98, 1.00 | 1.01 | 1.00, 1.02 | 1.00 | 0.99, 1.01 | 0.99 | 0.98, 1.01 |
| *Unemployed* | 0.92 | 0.88, 0.96 | 0.96 | 0.91, 1.00 | 0.79 | 0.76, 0.82 | 0.84 | 0.79, 0.89 |
| *Stayer* | 0.97 | 0.95, 0.99 | 1.02 | 1.00, 1.05 | 1.03 | 1.00, 1.05 | 1.01 | 0.98, 1.04 |
| *Changer* | 0.97 | 0.94, 1.00 | 1.02 | 0.99, 1.05 | 0.96 | 0.93, 0.98 | 1.01 | 0.98, 1.05 |
| *Adjusted*[c]: | | | | | | | | |
| *Not exposed* | 1.02 | 1.00, 1.03 | 1.06 | 1.05, 1.07 | 1.05 | 1.03, 1.06 | 1.06 | 1.04, 1.08 |
| *Unemployed* | 0.97 | 0.93, 1.01 | 1.04* | 0.99, 1.09 | 0.85* | 0.81, 0.89 | 0.95* | 0.89, 1.00 |
| *Stayer* | 1.00 | 0.98, 1.03 | 1.08 | 1.05, 1.11 | 1.09* | 1.06, 1.11 | 1.10 | 1.07, 1.13 |
| *Changer* | 1.00 | 0.97, 1.03 | 1.08* | 1.05, 1.11 | 1.01* | 0.99, 1.04 | 1.11* | 1.07, 1.15 |

[a] * Is used as an indication for a period where the exposed group differ from the unexposed according to contrasting odds ratio test

[b] Crude model, only adjusted for calendar year

[c] Adjusted model, adjustment for calendar year, sociodemographic factors and previous sickness absence

[d] The prevalence at time -3 is compared to prevalence at time -4 before downsizing

[e] The prevalence at time -1 is compared to prevalence at time -3 before downsizing

[f] The prevalence at time +1 after downsizing is compared to prevalence at time -1 before downsizing

[g] The prevalence at time +4 is compared to prevalence at time +1 after downsizing

demographics and previous sickness absence. As in the main analysis, compared to the unexposed, unemployed increased the number of purchases of anxiolytics during late pre-downsizing but then decreased to the end of follow up. The stayers continued to increase their number of purchases in the peri- (Rate Ratio (RR) 1.08, 95% CI 1.04, 1.13) and the post-downsizing period (RR 1.09, 95% CI 1.03, 1.15) more than the unexposed, see S1 Table. A similar pattern was observed for stayers and unemployed when looking at the number of purchased sedatives, while the changers seemed to increase more than the unexposed during peri- and post downsizing ('+4 vs -1'), see S2 Fig.

Stratified analysis was performed by sex, showing similar patterns for all psychotropic drugs for men and women (see S3 and S4 Figs). Further, we stratified the analysis by age group yielding similar patterns across age groups and drug type, although the prevalence from the peri-downsizing phase and onwards declined more in the youngest age group (S5 and S6 Figs).

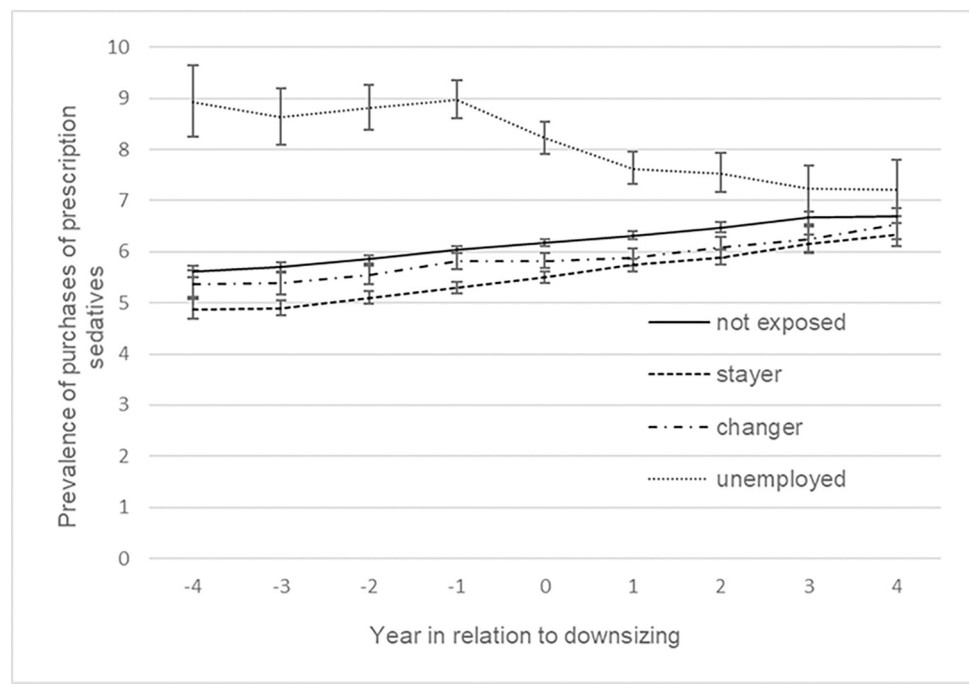

**Fig 3. Prevalence of purchased prescribed sedatives by employment status (%).** Adjusted for sociodemographic factors and previous sickness absence, with 95% CI.

## Purchases of prescribed psychotropic drugs in relation to a downsizing event taking previous sickness absence into account

In Table 3, results from adjusted analysis are presented, stratified by previous sickness absence. All groups with previous sickness absence increased their prevalence of purchases of anxiolytics before the event, see Fig 4. Moreover, unemployed (OR 1.25, 95% CI 1.17, 1.34) increased their purchases late pre-downsizing more than the non-exposed (OR. 1.18 95% CI 1.15, 1.21). During the peri-downsizing period, and post-downsizing ('+4 vs -1') the unemployed decreased their prevalence of purchasing anxiolytics more compared to the unexposed. Although the stayers and changers increased their odds of any purchases during the late pre-downsizing period, see Fig 4, this increase did not differ from the unexposed. In contrast to the unexposed, changers did not decrease their purchases of anxiolytics post-downsizing. For average number of purchases of anxiolytics all the exposed groups increased more than the unexposed during the late pre-downsizing phase (S7 Fig).

Looking at the corresponding figures for people without previous sickness absence (Fig 5 and S8 Fig), all groups had a lower prevalence of purchasing prescribed anxiolytics before the event followed by a slight increase from one year before the event and onwards. In the early pre-downsizing period the unemployed had a larger decrease than the unexposed. Stayers increased their odds of purchasing prescribed anxiolytics somewhat more than the unexposed from one year before the event up to four years after while changes among unemployed and changers did not differ from that of unexposed according to the contrast test.

A similar pattern was observed when studying purchases of sedatives among people with and without previous sickness absence, see S9–S12 Figs. Results for stayers and changers with a prior sickness absence showed a stronger increase of sedatives during the late pre-downsizing phase compared to the unexposed, whereas the unemployed had less of an increase. More pronounced decreasing trends for changers and unemployed were found during the peri-

**Table 3. Odds ratios (OR) for purchases of prescription anxiolytics and sedatives and their 95% confidence interval (CI) in relation to the downsizing event, comparing changes over time within groups in analyses stratified by downsizing and employment status after downsizing, and previous sickness absence.** Any statistically significant differences between exposed and not exposed are expressed with * in mutually adjusted models.

| | Early pre-downsizing '-3 vs -4'c | | Late pre-downsizing '-1 vs -3'd | | Peri-downsizing '+1 vs -1'e | | Post-downsizing '+4 vs +1'f | | '+4 vs -1'g |
|---|---|---|---|---|---|---|---|---|---|
| | OR | 95% CI | OR | 95% CI | OR | 95% CI | OR | 95% CI | |
| **Anxiolyticsb)** | | | | | | | | | |
| _Previous sickness absence_ | | | | | | | | | |
| _Not exposed_ | 1.09 | 1.07, 1.12 | 1.18 | 1.15, 1.21 | 0.80 | 0.78, 0.82 | 0.94 | 0.91, 0.96 | |
| _Unemployed_ | 1.05 | 0.98, 1.12 | 1.25* | 1.17, 1.34 | 0.73* | 0.69, 0.78 | 0.93 | 0.85, 1.01 | * |
| _Stayer_ | 1.10 | 1.05, 1.15 | 1.26 | 1.20, 1.31 | 0.81 | 0.78, 0.85 | 0.97 | 0.92, 1.02 | |
| _Changer_ | 1.11 | 1.05, 1.18 | 1.25 | 1.18, 1.32 | 0.77 | 0.74, 0.82 | 1.01* | 0.94, 1.08 | |
| _NO previous sickness absence_ | | | | | | | | | |
| _Not exposed_ | 0.93 | 0.91, 0.95 | 0.87 | 0.85, 0.88 | 1.25 | 1.22, 1.27 | 1.04 | 1.02, 1.07 | |
| _Unemployed_ | 0.80* | 0.74, 0.88 | 0.81 | 0.73, 0.88 | 1.09 | 1.00, 1.18 | 0.89 | 0.81, 0.99 | |
| _Stayer_ | 0.95 | 0.92, 0.99 | 0.88 | 0.85, 0.91 | 1.27 | 1.23, 1.31 | 1.09 | 1.05, 1.13 | * |
| _Changer_ | 0.94 | 0.90, 0.99 | 0.87 | 0.83, 0.92 | 1.21 | 1.16, 1.27 | 1.05 | 1.00, 1.10 | |
| **Sedatives b)** | | | | | | | | | |
| _Previous sickness absence_ | | | | | | | | | |
| _Not exposed_ | 1.11 | 1.09, 1.12 | 1.24 | 1.21, 1.26 | 0.85 | 0.84, 0.87 | 0.99 | 0.97, 1.02 | |
| _Unemployed_ | 1.04 | 0.99, 1.10 | 1.21* | 1.14, 1.28 | 0.78* | 0.74, 0.82 | 0.92* | 0.85, 0.99 | * |
| _Stayer_ | 1.09 | 1.05, 1.13 | 1.32* | 1.28, 1.38 | 0.86 | 0.83, 0.89 | 1.02 | 0.97, 1.07 | |
| _Changer_ | 1.12 | 1.07, 1.17 | 1.33* | 1.27, 1.40 | 0.79* | 0.76, 0.83 | 1.05* | 0.99, 1.11 | |
| _NO previous sickness absence_ | | | | | | | | | |
| _Not exposed_ | 0.95 | 0.94, 0.97 | 0.93 | 0.91, 0.94 | 1.22 | 1.20, 1.24 | 1.11 | 1.09, 1.13 | |
| _Unemployed_ | 0.85 | 0.79, 0.91 | 0.76 | 0.70, 0.82 | 1.00* | 0.92, 1.08 | 1.00 | 0.91, 1.10 | |
| _Stayer_ | 0.95 | 0.92, 0.98 | 0.92 | 0.89, 0.95 | 1.28* | 1.24, 1.31 | 1.16 | 1.12, 1.20 | * |
| _Changer_ | 0.93 | 0.90, 0.97 | 0.91 | 0.88, 0.95 | 1.21 | 1.17, 1.26 | 1.16* | 1.11, 1.21 | |

[a] a as an indication for a period where the exposed group differ from the unexposed according to contrasting odds ratio test

[b] Adjusted model, adjustment for calendar year, sociodemographic factors and previous sickness absence

[c] The prevalence at time -3 is compared to prevalence at time -4 before downsizing

[d] The prevalence at time -1 is compared to prevalence at time -3 before downsizing

[e] The prevalence at time +1 after downsizing is compared to prevalence at time -1 before downsizing

[f] The prevalence at time +4 is compared to prevalence at time +1 after downsizing

[g] The prevalence at time +4 after downsizing is compared to prevalence at time -1 before downsizing

downsizing phase, than for the unexposed. Changers with previous sickness absence increased their purchases more than the unexposed during the post-downsizing period, whereas the unemployed decreased more in this period.

Stayers without previous sickness absence increased their purchases of sedatives more than the unexposed from one year before downsizing and onwards (+4 vs -1) and in particular the peri-downsizing phase. Changers increased their prevalence more than the unexposed post downsizing.

## Discussion

We found a significant, albeit weak association between exposure to downsizing and prevalence of anxiolytic and sedative drugs. The results indicated that the use of these psychotropic drugs increased most for stayers during and after downsizing.

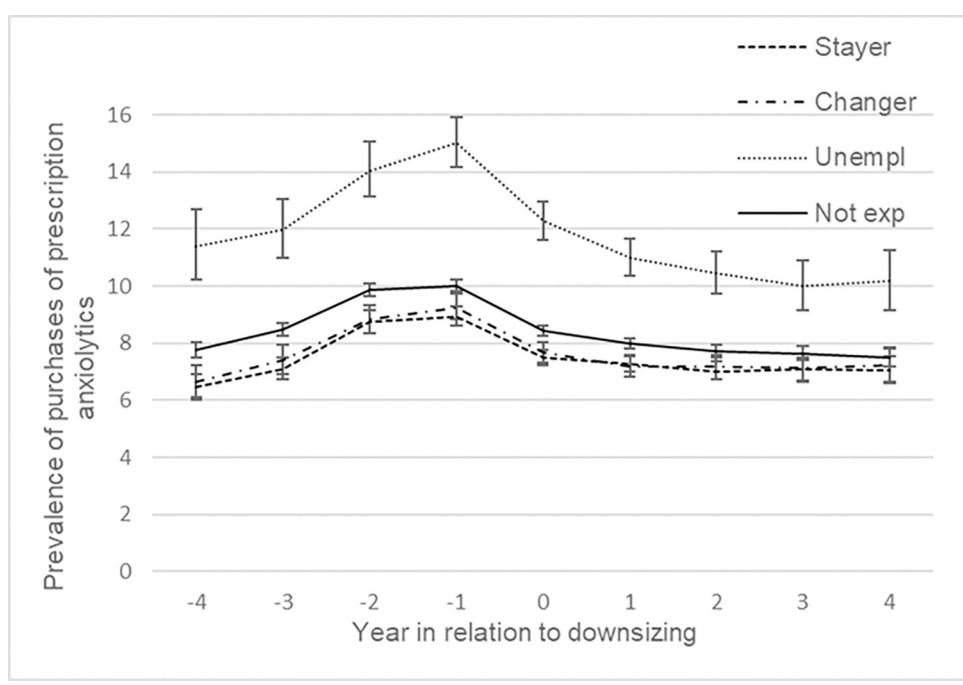

**Fig 4. Prevalence of purchased prescribed anxiolytics among people with a previous sickness absence.** Adjusted for sociodemographic factors and with 95% CI.

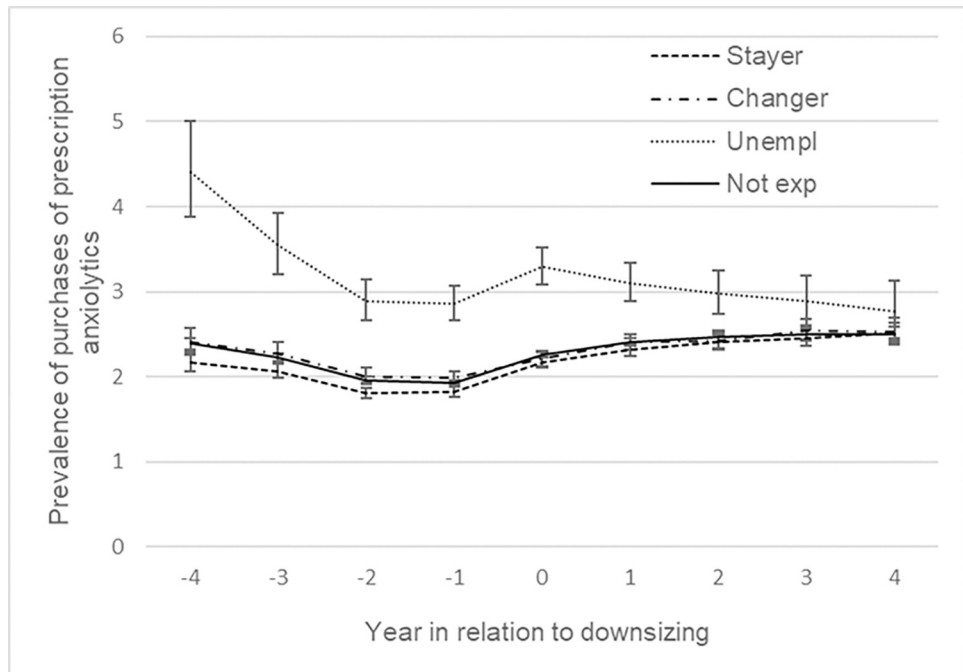

**Fig 5. Prevalence of purchased prescribed anxiolytics among people without a previous sickness absence.** Adjusted for sociodemographic factors and with 95% CI.

The different time trends in purchases of psychotropic drugs between exposed and unexposed employees, during certain phases, indicated that downsizing might entail treatment of mental health problems. During the period prior to the downsizing event, often referred to as the anticipation phase, it was primarily the unemployed and the changer groups that seemed to experience an increase in psychotropic drug purchases. The findings are in accordance with previous studies showing that the anticipation phase before a job change or unemployment has predominantly detrimental health effects [29–31]. This phase might be characterized by high levels of uncertainty about the future causing stress reactions, poor self-rated health, poor psychological wellbeing and emotional exhaustion [30, 32–34]. In addition, previous studies have shown that the time before an event is characterized by poorer perceived work environment in terms of work load and relations to co-workers and managers [3, 33, 35].

While the prevalence of purchasing psychotropic drugs decreased among the unemployed it increased among those who stayed within the organization after a downsizing. This could be a result of an increased work load [33], and feelings of guilt over colleagues who were laid off [8]. Downsizing may also entail a pressure to modify jobs in ways that lead to lowered skill discretion, decreased decision authority and participation in decision making, and reduced work predictability [36]. The decreasing pattern among the unemployed from the peri-downsizing period and onwards might reflect a situation characterized by relief from the major stressor of uncertainty [37]. It might also reflect less need for sedatives and anxiolytics in order to cope and perform at work.

According to a previous study which followed employees in Sweden from two years before to two years after an organizational downsizing, employees exposed to downsizing had higher odds of purchasing prescribed antidepressant compared to employees unexposed to downsizing [10]. Adding to the results of that study, we also found that during the pre-downsizing period the changers and unemployed increased their purchases of psychotropics in comparison to unexposed. However, while we observed a marked drop in sedative and anxiolytic purchases during the peri-period particularly among the unemployed, when compared to the unexposed, in the study by Magnusson et al. [10] the trends in antidepressant purchases for changers and unemployed during this period were more similar to that of the unexposed. The limited differences across the studies could be influenced by the high level of comorbidity between anxiety, sleep problems and depressive symptoms, the multiple indications that can exist for a certain type of psychotropic medication, and further that anxiolytics and sedatives can be used as pro re nata medications for insomnia, agitation or anxiety while treatments for depression are recommended to be more long-term. Sedatives and anxiolytics purchases may therefore capture some of the more subthreshold symptoms of poor mental health which may not have been captured in the study by Magnusson et al. The results further underscore the necessity of considering the various employment situations that employees may experience in connection with a downsizing, which in turn may impact differently on their mental health.

In order to address the possibility that the results were explained by health selection we stratified our sample by previous sickness absence. Bernstrom [38] studied whether stages of job change could be reflected in sickness absence and found that two years before leaving an organization the odds of sickness absence increased and then dropped again when the subjects entered a new organization. Our results on psychotropic drugs show a similar pattern in most cases, suggesting that the period before leaving one's job is detrimental in terms of feelings of anxiety and sleeping problems. After stratification by sickness absence during the two years before the event the stayers and changers with previous sickness absence had a higher increase in prevalence of purchasing sedatives compared to the unexposed group during the years before the event. The same was observed for anxiolytics, but only among the unemployed. Among those without previous sickness absence, stayers increased their prevalence during the

peri- and post-downsizing period compared to the unexposed group. Thus even when taking previous sickness absence into account, being exposed to a downsizing and subsequently remain in paid work was associated with a higher increase in prevalence of purchasing psychotropic drugs compared to not being exposed.

## Strengths and limitations

This study is based on a total population of employed Swedes with virtually no missing data, thus the sample is highly representative of the Swedish work force. The Swedish National Prescribed Drug Register contains information on all prescribed drugs dispensed at a Swedish pharmacy and thus has a good coverage of purchases of the drugs in focus in the present study. Information on redeemed prescribed sedatives and anxiolytics from several years before and after exposure to downsizing were utilized, making it possible to capture changes within different phases of the downsizing process. By examining different types of drugs this study can also contribute to a better understanding of potential short- and long-term effects of downsizing.

Due to the long period with available data we were able to utilize exposure to downsizing occurring in four different 12-month periods, making the results less sensitive to conditions of the labor market in one specific year. Hence, the associations between downsizing and purchases of psychotropic drugs are not likely to be influenced solely by the recession in 2008, although more major downsizing events occurred shortly after the economic downturn. Furthermore, dividing the exposed group into different categories based on their employment situation the year after downsizing made it possible to investigate the heterogeneity in purchases among different exposed groups. Since information on exposure and outcome is based on register data, this study should furthermore not be influenced by, e.g., recall bias and common method bias.

However, using only register based information on downsizing infers a risk of misclassification. From the Statistics on Dynamics of Enterprises and Establishments (DEE) register we only know whether a workplace reduced their staff or not between two subsequent years but not whether the person we defined as being exposed to downsizing was directly affected by staff reductions at that workplace. The downsizing could have been restricted to specific units at the workplace. On the contrary there might be people who have been classified as non-exposed although they might work in a unit heavily targeted with major staff reductions. Employees classified as unexposed could also have been exposed to minor downsizing. This type of misclassification would result in an underestimation of any effect of downsizing on mental health. Moreover, we have used the most commonly applied definition of major downsizing, but according to our knowledge, a standard definition of major downsizing is lacking. Although previous studies have shown stronger associations between major downsizing and health than for minor downsizing [1], further work may be needed on degree of downsizing and health effects.

Regarding purchases of prescribed sedatives and anxiolytics, a couple of limitations are worth noting. Studying the number of purchases made is somewhat crude as it does not capture the variation in dose-levels, meaning that differences in defined daily dosages are not detected. However, results regarding the number of purchases per person did not differ much from the trends of any purchases made and the problem of non-adherence would persist even if a more detailed measure was applied. Furthermore, the people captured in this study might be what Last and Adelaide call the tip of the iceberg [39]. Not all people with sleep disorders or anxiety will seek health care and not all prescribed medication will be redeemed. We also know that health care utilization in relation to need varies between socioeconomic groups [40], and gender differences in reporting and handling mental disorders has been observed

[41, 42]. However, health care in Sweden is largely tax-funded providing relatively equal access to health-care services irrespective of employment status.

## Conclusion

To conclude, this study supports an association between downsizing and purchases of prescribed psychotropic drugs. The results indicating an increase in psychotropic drug purchases prior to the downsizing event among changes and those who became unemployed after the downsizing implies that the anticipation-phase is a stressful time in the downsizing process for the exposed employees. The results however, suggested an even more prominent increase psychotropic drug purchases across the period of downsizing and the following years among those who continued to work in the same workplace, i.e., the stayers. This also implies that an organizational downsizing can have long-term consequences on mental health for those who stay in the organization, which did not appear to be attributed to health selection, indicating a need for preventive measures.

## Supporting information

**S1 Appendix. Analysis script for the GEE method.**
(DOCX)

**S1 Fig. Average number of purchases of anxiolytics by employment status after downsizing.** Adjusted for sociodemographic factors and previous sickness absence, with 95% CI.
(DOCX)

**S2 Fig. Average number of purchases of sedatives by employment status after downsizing.** Adjusted for sociodemographic factors and previous sickness absence, with 95% CI.
(DOCX)

**S3 Fig. Prevalence of purchased prescribed sedatives by sex (%).** Adjusted for sociodemographic factors and previous sickness absence.
(DOCX)

**S4 Fig. Prevalence of purchased prescribed anxiolytics by sex (%).** Adjusted for sociodemographic factors and previous sickness absence.
(DOCX)

**S5 Fig. Prevalence of purchased prescribed sedatives by age groups (%).** Adjusted for sociodemographic factors and previous sickness absence.
(DOCX)

**S6 Fig. Prevalence of purchased prescribed anxiolytics by age groups (%).** Adjusted for sociodemographic factors and previous sickness absence.
(DOCX)

**S7 Fig. Average number of purchases of anxiolytics by employment status after downsizing, among individuals with prior sick leave.** Adjusted for sociodemographic factors, with 95% CI.
(DOCX)

**S8 Fig. Average number of purchases of anxiolytics by employment status after downsizing, among individuals without prior sick leave.** Adjusted for sociodemographic factors, with 95% CI.
(DOCX)

**S9 Fig. Prevalence of purchased prescribed sedatives by employment status after downsizing, among individuals with prior sick leave.** Adjusted for sociodemographic factors, with 95% CI.
(DOCX)

**S10 Fig. Average number of purchases of sedatives by employment status after downsizing, among individuals with prior sick leave.** Adjusted for sociodemographic factors, with 95% CI.
(DOCX)

**S11 Fig. Prevalence of purchased prescribed sedatives by employment status after downsizing, among individuals without prior sick leave.** Adjusted for sociodemographic factors, with 95% CI.
(DOCX)

**S12 Fig. Average number of purchases of sedatives by employment status after downsizing, among individuals without prior sick leave.** Adjusted for sociodemographic factors, with 95% CI.
(DOCX)

**S1 Table. Rate ratios (RR) for purchases of prescription anxiolytics and sedatives and their 95% confidence interval (CI) in relation to a downsizing event, comparing changes over time within groups in analyses stratified by downsizing and employment status after downsizing.** Statistically significant differences between exposed and not exposed are expressed with * in mutually adjusted models.
(DOCX)

## Acknowledgments

The authors want to thank Jaana Pentti, statistician at the Department of Public Health University of Turku, for valuable contributions on statistical analysis and coding. We also thank Tatjana von Rosen at the Department of Statistics, Stockholm University, for her comments on data management and coding of key variables. The authors are grateful to the Swedish Research Council for Health, Working Life and Welfare (FORTE) and colleagues at the Stress Research Institute and Division of Insurance Medicine at Karolinska Institute for enabling this work.

## Author Contributions

**Conceptualization:** Sandra Blomqvist, Kristina Alexanderson, Jussi Vahtera, Hugo Westerlund, Linda L. Magnusson Hanson.

**Data curation:** Sandra Blomqvist, Linda L. Magnusson Hanson.

**Formal analysis:** Sandra Blomqvist, Jussi Vahtera, Hugo Westerlund, Linda L. Magnusson Hanson.

**Funding acquisition:** Kristina Alexanderson, Hugo Westerlund, Linda L. Magnusson Hanson.

**Methodology:** Kristina Alexanderson, Jussi Vahtera, Hugo Westerlund, Linda L. Magnusson Hanson.

**Software:** Jussi Vahtera.

**Supervision:** Hugo Westerlund, Linda L. Magnusson Hanson.

**Validation:** Kristina Alexanderson, Jussi Vahtera, Hugo Westerlund, Linda L. Magnusson Hanson.

**Writing – original draft:** Sandra Blomqvist.

**Writing – review & editing:** Sandra Blomqvist, Kristina Alexanderson, Jussi Vahtera, Hugo Westerlund, Linda L. Magnusson Hanson.

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
