## [Decision Letter · Decision Letter 0]

11 Apr 2023

PONE-D-23-00866Downsizing and purchases of psychotropic drugs: a longitudinal study of stayers, changers and unemployedPLOS ONE

Dear Dr. Blomqvist,

Thank you for submitting your manuscript to PLOS ONE. After careful consideration, we feel that it has merit but does not fully meet PLOS ONE’s publication criteria as it currently stands. Therefore, we invite you to submit a revised version of the manuscript that addresses the points raised during the review process.

We look forward to receiving your revised manuscript.

Kind regards,

Ali B. Mahmoud, Ph.D.

Academic Editor

PLOS ONE

Journal Requirements:

https://journals.plos.org/plosone/s/file?id=ba62/PLOSOne_formatting_sample_title_authors_affiliations.pdf"

3. Please amend the manuscript submission data (via Edit Submission) to include author Alexanderson Kristina, Vahtera Jussi, Westerlund Hugo and Magnusson Hanson L. Linda.

Reviewers' comments:

Reviewer's Responses to Questions

**Comments to the Author**

1. Is the manuscript technically sound, and do the data support the conclusions?

Reviewer #1: Yes

Reviewer #2: Yes

2. Has the statistical analysis been performed appropriately and rigorously? 

Reviewer #1: Yes

Reviewer #2: Yes

3. Have the authors made all data underlying the findings in their manuscript fully available?

Reviewer #1: Yes

Reviewer #2: No

4. Is the manuscript presented in an intelligible fashion and written in standard English?

Reviewer #1: Yes

Reviewer #2: Yes

5. Review Comments to the Author

Reviewer #1: This is well written descriptive evidence that the study indicates that being exposed to downsizing is associated with increased use of sedatives and anxiolytics, before the event but especially thereafter for employees who stay in the organization. The major statistical approach is the use of the GEE with applied repeated-measures logistic regression analysis (OR and 95% CI).The most informative presentations are Table 3, trajectory Figures 2 to 5 and supplemental figures 1 to 8. The sample size is certainly adequate with the four major participating groups stratified accordingly.

As noted above, the trends are described by the investigators. There is little or no inferential statistics except for the confidence intervals implying two sided statistical significance as seen in the text and in Table 3.

Reviewer #2: This is an interesting and well-written paper on an important topic.

1) There is a special story about the paper: The paper is actually already published at Plos One (Reference #17) but was, according to the authors, retracted due to a coding mistake. Thus, this paper is the corrected version of the earlier paper, reference #17, that has been retracted. The authors describe the coding error and that this paper is a corrected version of the earlier paper in the end of the introduction. However, it does not say in the introduction that the earlier version was retracted, one only becomes aware of this, when looking up the reference in the reference list. Please make it clear, also in the introduction, that the previous paper was retracted.

2) Further, I suggest in the interest of full transparency, that you address both in the abstract and in the title that this is a corrected version of a previous paper that had been retracted. For example to change the title to: “Downsizing and purchases of psychotropic drugs: a longitudinal study of stayers, changers and unemployed. Corrected version of a retracted paper”. This is important, because right now the title of the retracted paper and the current paper is identical, which could be a source of confusion.

3) Strangely, when I looked up the previous paper in pubmed and on the Plos One Website, it does not say so that the paper was retracted. At least I could not find a notice on retraction there. Please clarify this.

4) I am uncertain how I should review this paper. As the previous paper went through peer-review and was accepted and published, and the current paper is a correction of a coding error, one could argue that the current paper should be automatically published after the coding error was corrected. However, since the journal sent me the paper and asked for a review, I did a normal review paper. But I am sure if this normal review is appropriate.

5) The seminal papers on downsizing and health of employees, who were affected by the downsizing in different ways (stayers, changers, unemployed) based on data from the great recession in Finland in the 1990s and published by Jussi Vahtera, Mika Kivimäki and others, need to be replicated in other settings (other countries and other time periods) and it is good that the current paper is doing this.

6) The first thing I was wondering about was why only anxiolytics and sedatives and not antidepressants? I agree, important to examine anxiolytics and sedatives but why did you leave out antidepressants? Was this because downsizing and antidepressants was already analyzed in the study by Magnusson Hanson et al 2016 (reference #10)? If this was the case, then please delineate in greater detail the results by Magnusson Hanson et al and compare them with your results. Because then, the whole picture of downsizing and use of psychotropic medication is distributed across two papers and it would be good if your paper synthesize the results from the two papers.

7) Why the cut-off point of >=18 for staff reduction? You write that this is “the most commonly used definition”, but is there any other reason why >=18 indicates major downsizing and staff reduction of 17% is not major? I suggest that you do sensitivity analyses with other cut-off points (e.g., >=13 or >=23) to examine if the results remain robust. It would be even better, if you could do dose-response analyses, comparing the risk of purchasing anxiolytics and sedatives for groups of no downsizing, minor downsizing and major downsizing.

8) With regard to supplementary analyses: Your outcome is prevalence of anxiolytics/sedatives purchase. Consider to analyze in addition incidence. Thus, to exclude everyone with a history of purchase before t minus 4 and then analyze incidence of purchase from t minus 4 to t plus 4.

9) With regard to the “changers”: Can you take into account, whether they changed to a job with a similar, higher or lower occupational grade?

10) With regard to “unemployed”. Can you take into account, how long the “unemployed” remained “unemployed” in subsequent years?

11) On page 16 you refer twice to “data not shown”. I suggest you show the data in Online Supplementary Material. For example, even though estimates were similar for men and women, showing the exact estimates stratified by sex would be informative, for example for future meta-analyses.

Minor comments:

12) Page 23: “we only know if a workplace reduced their staff”. I am not a native English speaker, but it is my feeling that this should be “we only know whether a workplace reduced their staff”.

12) Page 23, last paragraph. Two sentences in a row start with “However”. Consider to change this.

6. PLOS authors have the option to publish the peer review history of their article (what does this mean?). If published, this will include your full peer review and any attached files.

Reviewer #1: No

Reviewer #2: No

---

## [Author Response · Author response to Decision Letter 0]

1 Jun 2023

Dear Editor and Reviewers,

Thank you very much for taking the time to read and comment on our manuscript. As pointed out by one of the reviewers, this paper was first published in 2018, but an error was then discovered by the authors in 2020. Therefore, we promptly contacted PLOS One with new and corrected analyses together with a thorough description of the mistake and its implications for the results. We have repeatedly described both the error and its implication upon request from staff at PLOS One administrating this case. As Reviewer 2 also expresses, the paper has already undergone a peer review, based on which it was accepted. We have now tried to address the new reviewer comments where possible, but we have not changed or added any further analyses as we want this paper to resemble our original paper as much as possible, apart from the mistake. The rationale behind this is that the paper has already been cited and readers should easily be able to check what has been changed in this new and corrected version. With this in mind, please see our responses in connection to the reviewer’s comments below, and in the revised manuscript, which is enclosed both with tracked changes, and with all changes accepted. We sincerely hope that, after this long process of handling our case, you will find our reasoning and edits satisfying.

On behalf of all the authors,

Sandra Blomqvist

Reviewers' comments:

Reviewer's Responses to Questions

Comments to the Author

1. Is the manuscript technically sound, and do the data support the conclusions?

Reviewer #1: Yes

Reviewer #2: Yes

2. Has the statistical analysis been performed appropriately and rigorously? 

Reviewer #1: Yes

Reviewer #2: Yes

3. Have the authors made all data underlying the findings in their manuscript fully available?

Reviewer #1: Yes

Reviewer #2: No 

4. Is the manuscript presented in an intelligible fashion and written in standard English?

Reviewer #1: Yes

Reviewer #2: Yes

5. Review Comments to the Author

Reviewer #1: This is well written descriptive evidence that the study indicates that being exposed to downsizing is associated with increased use of sedatives and anxiolytics, before the event but especially thereafter for employees who stay in the organization. The major statistical approach is the use of the GEE with applied repeated-measures logistic regression analysis (OR and 95% CI).The most informative presentations are Table 3, trajectory Figures 2 to 5 and supplemental figures 1 to 8. The sample size is certainly adequate with the four major participating groups stratified accordingly.

As noted above, the trends are described by the investigators. There is little or no inferential statistics except for the confidence intervals implying two sided statistical significance as seen in the text and in Table 3.

Thank you.

Reviewer #2: This is an interesting and well-written paper on an important topic.

1) There is a special story about the paper: The paper is actually already published at Plos One (Reference #17) but was, according to the authors, retracted due to a coding mistake. Thus, this paper is the corrected version of the earlier paper, reference #17, that has been retracted. The authors describe the coding error and that this paper is a corrected version of the earlier paper in the end of the introduction. However, it does not say in the introduction that the earlier version was retracted, one only becomes aware of this, when looking up the reference in the reference list. Please make it clear, also in the introduction, that the previous paper was retracted.

Thank you for reading our paper and yes, the paper has a special story to it. Regarding the status of the former version, we have followed the guidelines that we received from PLOS One. The procedure that PLOS One decided on, was that the corrected version should undergo yet another review and that the old paper should be replaced by the new (along with a retraction notice) as soon as the review process was over, given that it wasn’t too long. However, the process for PLOS One to find suitable reviewers took long and they decided to retract it anyway, thus the paper changed status after this new version was submitted, while awaiting reviewers. The formulations used in the paper concerning the old manuscript were those that the Plos One ethics team instructed us to use. However, we now suggest a slight revision of the formulation by the ethics team about the paper being retracted. Please see p. 4 in the version with tracked changes.

The present study is the corrected version of a previously published version of this paper, now retracted, in which we found errors due to a coding mistake [17].”

2) Further, I suggest in the interest of full transparency, that you address both in the abstract and in the title that this is a corrected version of a previous paper that had been retracted. For example to change the title to: “Downsizing and purchases of psychotropic drugs: a longitudinal study of stayers, changers and unemployed. Corrected version of a retracted paper”. This is important, because right now the title of the retracted paper and the current paper is identical, which could be a source of confusion.

As mentioned above, the ethics team at PLOS One instructed us exactly how the process concerning the retraction and resubmission of this paper should be communicated to future readers. However, if PLOS One so decided, we would have no problem to add the information suggested by the reviewer to the title.

3) Strangely, when I looked up the previous paper in pubmed and on the Plos One Website, it does not say so that the paper was retracted. At least I could not find a notice on retraction there. Please clarify this.

We as authors neither fully understand nor agree with how our case have been handled. The paper was not retracted when this revised version was submitted even if we repeatedly pointed out to the journal that the paper was still available to read and cite, which of course was highly problematic. Therefore, we hope that the PLOS One ethics team, administrative team and/or editors can clarify this. 

4) I am uncertain how I should review this paper. As the previous paper went through peer-review and was accepted and published, and the current paper is a correction of a coding error, one could argue that the current paper should be automatically published after the coding error was corrected. However, since the journal sent me the paper and asked for a review, I did a normal review paper. But I am sure if this normal review is appropriate.

Thank you for spelling this out, and we agree. This manuscript is meant to be a corrected version of a previously peer reviewed paper and for transparency we believe that it should be minimally different from that paper as possible, apart from the mistake that we unfortunately made. 

5) The seminal papers on downsizing and health of employees, who were affected by the downsizing in different ways (stayers, changers, unemployed) based on data from the great recession in Finland in the 1990s and published by Jussi Vahtera, Mika Kivimäki and others, need to be replicated in other settings (other countries and other time periods) and it is good that the current paper is doing this.

Thank you, and we agree that this is important to do.

6) The first thing I was wondering about was why only anxiolytics and sedatives and not antidepressants? I agree, important to examine anxiolytics and sedatives but why did you leave out antidepressants? Was this because downsizing and antidepressants was already analyzed in the study by Magnusson Hanson et al 2016 (reference #10)? If this was the case, then please delineate in greater detail the results by Magnusson Hanson et al and compare them with your results. Because then, the whole picture of downsizing and use of psychotropic medication is distributed across two papers and it would be good if your paper synthesize the results from the two papers.

Yes, this was because antidepressants were studied by Magnusson et. al. in 2016, and our intention with the present study was to build on that to examine temporal changes of different mental health conditions such as sleep and anxiety across the downsizing process, which potentially could occur earlier or precede depression. This was expressed very briefly in the discussion section but has now been elaborated slightly, See p. 21 and below. 

According to a previous study which followed employees in Sweden from two years before to two years after an organizational downsizing, employees exposed to downsizing had higher odds of purchasing prescribed antidepressant compared to employees unexposed to downsizing. [10] Adding to the results of that study, we also found that during the pre-downsizing period the changers and unemployed increased their purchases of psychotropics in comparison to unexposed. However, while we observed a marked drop in sedative and anxiolytic purchases during the peri-period particularly among the unemployed, when compared to the unexposed, in the study by Magnusson et. al. [10] the trends in antidepressant purchases for changers and unemployed during this period were more similar to that of the unexposed. The limited differences across the studies could be influenced by the high level of comorbidity between anxiety, sleep problems and depressive symptoms, the multiple indications that can exist for a certain type of psychotropic medication, and further that anxiolytics and sedatives can be used as pro re nata medications for insomnia, agitation or anxiety while treatments for depression are recommended to be more long-term. Sedatives and anxiolytics purchases may therefore capture some of the more subthreshold symptoms of poor mental health which may not have been captured in the study by Magnusson et. al. The results further underscore the necessity of considering the various employment situations that employees may experience in connection with a downsizing, which in turn may impact differently on their mental health. 

7) Why the cut-off point of >=18 for staff reduction? You write that this is “the most commonly used definition”, but is there any other reason why >=18 indicates major downsizing and staff reduction of 17% is not major? I suggest that you do sensitivity analyses with other cut-off points (e.g., >=13 or >=23) to examine if the results remain robust. It would be even better, if you could do dose-response analyses, comparing the risk of purchasing anxiolytics and sedatives for groups of no downsizing, minor downsizing and major downsizing.

We use it because it has become a de facto standard definition since it was used in the seminal paper by Vahtera et al. as well as by Magnusson et. al. 2016, which the current study builds on. To our understanding the cut off is arbitrary and does not have a clear theoretical underpinning, at least not discussed by Vahtera et. al. (1997). Others, including Vahtera et. al and Magnusson et. al. 2016, have shown previously that the association with sick leave spells, and antidepressant purchases, respectively seem to be stronger for major downsizing than minor downsizing. We could of course also have done this type of analysis but chose to focus on others, such as moderation by a person’s prior health status, age, sex etc. We suggest the following addition to the discussion though, to better clarify to reader our choice of cut off for the exposure. See p. 23-24.

“Moreover, we have used the most commonly applied definition of major downsizing, but according to our knowledge, a standard definition of major downsizing is lacking. Although previous studies have shown stronger associations between major downsizing and health than for minor downsizing, [1] further work may be needed on degree of downsizing and health effects.”

8) With regard to supplementary analyses: Your outcome is prevalence of anxiolytics/sedatives purchase. Consider to analyze in addition incidence. Thus, to exclude everyone with a history of purchase before t minus 4 and then analyze incidence of purchase from t minus 4 to t plus 4.

Incident purchases of anxiolytics and sedatives could indeed have been interesting. However, the Swedish prescribed drug register which includes all redeemed psychotropic drug prescriptions, was introduced in July 2005 and thus prevent us from using psychotropic drug medications further back in time, at timepoint - 5 years or earlier, as a way of controlling for prior mental health problems. As an alternative, we chose to handle potential reversed causation by prior mental health problems by stratifying on sick leave spells occurring two years prior to (or earlier) the downsizing. Furthermore, the outcome we used was odds ratio for purchasing prescribed psychotropics in a certain 12-month period, but we also contrasted the odds in one 12-month period with the odds in an earlier 12-month period within each group, and further against the unexposed group. This means that we analyzed change in purchases over time, not only prevalences in specific time periods. In addition, we also studied change in amount of purchases across time, in the same manner. This approach allows us to understand temporal dynamics in a way that would not be possible if we would only analyze incident cases.

9) With regard to the “changers”: Can you take into account, whether they changed to a job with a similar, higher or lower occupational grade?

It could be possible but would require some fairly technical programming, and as it is slightly out of the scope for this paper, we feel that it is not motivated at this stage. 

10) With regard to “unemployed”. Can you take into account, how long the “unemployed” remained “unemployed” in subsequent years?

This is also possible and we did briefly look into how the length of the unemployment spell affected the trajectories of psychotropic drug purchase to better understand the group of unemployed. I recall that we found what others have presented before us, about job loss and mental health, namely that longer spells were associated with worse mental health than shorter spells. 

11) On page 16 you refer twice to “data not shown”. I suggest you show the data in Online Supplementary Material. For example, even though estimates were similar for men and women, showing the exact estimates stratified by sex would be informative, for example for future meta-analyses.

Thank you for this comment, we have now included the data as supplementary material instead and provide a note in the text where the reader can find it. Please see. p. 16.

“Stratified analysis was performed by sex, showing similar patterns for all psychotropic drugs for men and women (see S3 and S4 Figs).”

And p. 17

“For average number of purchases of anxiolytics all the exposed groups increased more than the unexposed during the late pre-downsizing phase (S7 Fig).”

Minor comments:

12) Page 23: “we only know if a workplace reduced their staff”. I am not a native English speaker, but it is my feeling that this should be “we only know whether a workplace reduced their staff”.

We have now changed to using “whether” instead of “if”.

12) Page 23, last paragraph. Two sentences in a row start with “However”. Consider to change this.

Thanks for noticing that, we have varied our wording to create a better flow for the reader. 

6. PLOS authors have the option to publish the peer review history of their article (what does this mean?). If published, this will include your full peer review and any attached files.

Do you want your identity to be public for this peer review? For information about this choice, including consent withdrawal, please see our Privacy Policy.

Reviewer #1: No

Reviewer #2: No

---

## [Decision Letter · Decision Letter 1]

1 Sep 2023

PONE-D-23-00866R1Downsizing and purchases of psychotropic drugs: a longitudinal study of stayers, changers and unemployedPLOS ONE

Dear Dr. Blomqvist,

Thank you for submitting your manuscript to PLOS ONE. After careful consideration, we feel that it has merit but does not fully meet PLOS ONE’s publication criteria as it currently stands. Therefore, we invite you to submit a revised version of the manuscript that addresses the points raised during the review process.

We look forward to receiving your revised manuscript.

Kind regards,

Ali B. Mahmoud, Ph.D.

Academic Editor

PLOS ONE

Journal Requirements:

1)  Citation of the retracted paper (Introduction and Reference list): The text addressing the retraction in the Introduction text is suitable, but the citation of the retraction and original publication is not correct. The in-text citations in the Introduction should refer to both the retracted/original article and the retraction notice, which should be listed as follows in the References list:

17. Blomqvist S, Alexanderson K, Vahtera J, Westerlund H, Magnusson Hanson LL (2018) Downsizing and purchases of psychotropic drugs: A longitudinal study of stayers, changers and unemployed. PLoS ONE 13(8): e0203433. https://doi.org/10.1371/journal.pone.0203433 (**Retracted**)

18. Blomqvist S, Alexanderson K, Vahtera J, Westerlund H, Magnusson Hanson LL (2023) Retraction: Downsizing and purchases of psychotropic drugs: A longitudinal study of stayers, changers and unemployed. PLOS ONE 18(4): e0285004. https://doi.org/10.1371/journal.pone.0285004

When correcting this, please also update the citations and referencing for the articles currently listed as reference numbers 18+.

2) Analysis scripts: please could the scripts used for the analyses in the study by provided either in a Supplementary Information file or via a public repository. 

Reviewers' comments:

Reviewer's Responses to Questions

**Comments to the Author**

1. If the authors have adequately addressed your comments raised in a previous round of review and you feel that this manuscript is now acceptable for publication, you may indicate that here to bypass the “Comments to the Author” section, enter your conflict of interest statement in the “Confidential to Editor” section, and submit your "Accept" recommendation.

Reviewer #2: All comments have been addressed

Reviewer #3: All comments have been addressed

2. Is the manuscript technically sound, and do the data support the conclusions?

Reviewer #2: Yes

Reviewer #3: Yes

3. Has the statistical analysis been performed appropriately and rigorously? 

Reviewer #2: Yes

Reviewer #3: Yes

4. Have the authors made all data underlying the findings in their manuscript fully available?

Reviewer #2: No

Reviewer #3: No

5. Is the manuscript presented in an intelligible fashion and written in standard English?

Reviewer #2: Yes

Reviewer #3: Yes

6. Review Comments to the Author

Reviewer #2: Dear Authors,

Thank you for your response and for considering some of my comments. I understand that you could not consider all comments, because this paper is a special case (with its retraction history) and that you had been instructed by Plos One how to handle this. I do not have further comments or requests.

Reviewer #3: Thank you for this opportunity to revisit your article concerning mental health and workplace downsizing.

I agree with what both the authors and the reviewers stated in R1: This is a difficult paper to review as it has already undergone the process once. Also, I tend to agree with the authors that as the original paper with the error in it has already been cited, it would be best to refrain from making large changes to the paper. To more complicate things, I did not participate in the first round of peer reviews and most of the points I would have raised and were already made by the two reviewers participating in the first round and then adequately addressed by the authors. I will attempt to review the paper as requested, but with the previous round in mind.

I think the authors have done a good job of describing the error in the original publication. I agree with what prior reviewers already suggested concerning the title, as it would be clearer to the reader if the retracted paper and the current version had separate titles. But this appears to be concern for the PLOS one team and not the authors.

Several other points were also made in the previous round of reviews which I believe the authors have now adequately addressed, eg. that concerning the inclusion of antidepressants and the cut offs for workplace downsizing. I would have suggested to present results for both anxiolytics/sedatives and antidepressants from one set of data and analyses (addressed both together and separately), as well as running the models for both anxiolytics and sedatives together. However, the authors make a good case for their choice to to adhere to the original analyses.

Minor concerns

Table 3. The title is: "Odds Ratios (OR) for purchases of prescription anxiolytics and sedatives and their 95% Confidence Interval (CI) in relation to the downsizing event, comparing changes over time within groups in analyses stratified by downsizing and employment status after downsizing. Any statistically significant differences between exposed and not exposed are expressed with * in mutually adjusted models. All analysis are

stratified by previous sickness absence." Would it be clearer if this was: "Odds Ratios (OR) for purchases of prescription anxiolytics and sedatives and their 95% Confidence Interval (CI) in relation to the downsizing event, comparing changes over time within groups in analyses stratified by downsizing and employment status after downsizing, and previous sickness absence. Any statistically significant differences between exposed and not exposed are expressed with * in mutually adjusted models."

Discussion. The paper would have benefited from a brief discussion on the public health or clinical relevance of the results. What are the implications of the somewhat modest associations found here? But as you have already made multiple changes following reviewers suggestions and as the space is limited, I am happy with the discussion as is.

In conclusions you write: "To conclude, this study supports an association between downsizing and purchases of

prescribed psychotropic drugs. The association was observed before the event but especially

prominent --" I'm not a native speaker, but I struggled with this section. Would something to the following effect be clearer to the reader: "This study supports an association between downsizing and purchases of

prescribed psychotropic drugs. The association was especially prominent for the stayers before the event, and they also increased their purchases during and after downsizing. --- further implies that organizational downsizing can have long term mental health consequences for those who stay in the organization, indicating a need for health promoting measures."

Altogether in the discussion, conclusions and abstract I sometimes found it difficult to say what your main conclusions for the leavers and the unemployed were. Perhaps these two groups could be included the conclusions, as well as add more detail on the main findings in the beginning of the discussion or the abstract?

7. PLOS authors have the option to publish the peer review history of their article (what does this mean?). If published, this will include your full peer review and any attached files.

Reviewer #2: No

Reviewer #3: No

---

## [Author Response · Author response to Decision Letter 1]

18 Oct 2023

Journal Requirements:

1) Citation of the retracted paper (Introduction and Reference list): The text addressing the retraction in the Introduction text is suitable, but the citation of the retraction and original publication is not correct. The in-text citations in the Introduction should refer to both the retracted/original article and the retraction notice, which should be listed as follows in the References list:

17. Blomqvist S, Alexanderson K, Vahtera J, Westerlund H, Magnusson Hanson LL (2018) Downsizing and purchases of psychotropic drugs: A longitudinal study of stayers, changers and unemployed. PLoS ONE 13(8): e0203433. https://doi.org/10.1371/journal.pone.0203433 (Retracted)

18. Blomqvist S, Alexanderson K, Vahtera J, Westerlund H, Magnusson Hanson LL (2023) Retraction: Downsizing and purchases of psychotropic drugs: A longitudinal study of stayers, changers and unemployed. PLOS ONE 18(4): e0285004. https://doi.org/10.1371/journal.pone.0285004

When correcting this, please also update the citations and referencing for the articles currently listed as reference numbers 18+.

Thank you for the clarification of how to correctly cite the different versions. We have now updated the manuscript according to the requirements and further updated the subsequent refences in the list after adding reference no. 18, see. p. 4 in the introduction and the reference list on p. 27, (manuscript version w track changes)

2) Analysis scripts: please could the scripts used for the analyses in the study by provided either in a Supplementary Information file or via a public repository. 

We have now included the analysis scripts as Supplementary Information, see file: 

S1 Appendix. Analysis script for GEE methods

And the following note included in the method section on p. 9:

“See S1 Appendix for the SAS 9.4 scripts used for these analyses.”

Reviewers' comments:

Reviewer's Responses to Questions

Comments to the Author

1. If the authors have adequately addressed your comments raised in a previous round of review and you feel that this manuscript is now acceptable for publication, you may indicate that here to bypass the “Comments to the Author” section, enter your conflict of interest statement in the “Confidential to Editor” section, and submit your "Accept" recommendation.

Reviewer #2: All comments have been addressed

Reviewer #3: All comments have been addressed

2. Is the manuscript technically sound, and do the data support the conclusions?

Reviewer #2: Yes

Reviewer #3: Yes

3. Has the statistical analysis been performed appropriately and rigorously? 

Reviewer #2: Yes

Reviewer #3: Yes

4. Have the authors made all data underlying the findings in their manuscript fully available?

Reviewer #2: No

Reviewer #3: No

5. Is the manuscript presented in an intelligible fashion and written in standard English?

Reviewer #2: Yes

Reviewer #3: Yes

6. Review Comments to the Author

Reviewer #2: Dear Authors,

Thank you for your response and for considering some of my comments. I understand that you could not consider all comments, because this paper is a special case (with its retraction history) and that you had been instructed by Plos One how to handle this. I do not have further comments or requests.

Thank you for taking the time to read and review our manuscript. 

Reviewer #3: Thank you for this opportunity to revisit your article concerning mental health and workplace downsizing.

I agree with what both the authors and the reviewers stated in R1: This is a difficult paper to review as it has already undergone the process once. Also, I tend to agree with the authors that as the original paper with the error in it has already been cited, it would be best to refrain from making large changes to the paper. To more complicate things, I did not participate in the first round of peer reviews and most of the points I would have raised and were already made by the two reviewers participating in the first round and then adequately addressed by the authors. I will attempt to review the paper as requested, but with the previous round in mind.

I think the authors have done a good job of describing the error in the original publication. I agree with what prior reviewers already suggested concerning the title, as it would be clearer to the reader if the retracted paper and the current version had separate titles. But this appears to be concern for the PLOS one team and not the authors.

Thank you and we would also like to refer this issue about the title to the PLOS one team and how titles for retraction and resubmissions are normally dealt with. 

Several other points were also made in the previous round of reviews which I believe the authors have now adequately addressed, eg. that concerning the inclusion of antidepressants and the cut offs for workplace downsizing. I would have suggested to present results for both anxiolytics/sedatives and antidepressants from one set of data and analyses (addressed both together and separately), as well as running the models for both anxiolytics and sedatives together. However, the authors make a good case for their choice to to adhere to the original analyses.

Yes, we acknowledge that improvement of the paper can be done and these are all good suggestions. However, as noted already, for this particular case we would like to stay as close to the original version as possible and not introduce new types of analysis since the old version has already been cited. 

Minor concerns

Table 3. The title is: "Odds Ratios (OR) for purchases of prescription anxiolytics and sedatives and their 95% Confidence Interval (CI) in relation to the downsizing event, comparing changes over time within groups in analyses stratified by downsizing and employment status after downsizing. Any statistically significant differences between exposed and not exposed are expressed with * in mutually adjusted models. All analysis are

stratified by previous sickness absence." Would it be clearer if this was: "Odds Ratios (OR) for purchases of prescription anxiolytics and sedatives and their 95% Confidence Interval (CI) in relation to the downsizing event, comparing changes over time within groups in analyses stratified by downsizing and employment status after downsizing, and previous sickness absence. Any statistically significant differences between exposed and not exposed are expressed with * in mutually adjusted models."

Yes, thank you, this reads a lot smoother. We have now updated the title according to your suggestion. See page 18, manuscript version w track changes.

Discussion. The paper would have benefited from a brief discussion on the public health or clinical relevance of the results. What are the implications of the somewhat modest associations found here? But as you have already made multiple changes following reviewers suggestions and as the space is limited, I am happy with the discussion as is.

In conclusions you write: "To conclude, this study supports an association between downsizing and purchases of

prescribed psychotropic drugs. The association was observed before the event but especially

prominent --" I'm not a native speaker, but I struggled with this section. Would something to the following effect be clearer to the reader: "This study supports an association between downsizing and purchases of

prescribed psychotropic drugs. The association was especially prominent for the stayers before the event, and they also increased their purchases during and after downsizing. --- further implies that organizational downsizing can have long term mental health consequences for those who stay in the organization, indicating a need for health promoting measures."

We see your point but we also believe that the suggestion points in a slightly different direction than what we have intended. The text alteration seems to suggest that stayers were particularly affected prior to the event but this was not the case for this group, they were rather affected during and after the event- although some effect seemed to exist prior to the event too (mostly for those with a previous sickness absence). However, our original formulation was not optimal either. We therefore suggest that it is worded in following way instead (on page 24-25 in the manuscript w tracked changes):

“To conclude, this study supports an association between downsizing and purchases of prescribed psychotropic drugs. The results indicating an increase in psychotropic drug purchases prior to the downsizing event among changes and those who became unemployed after the downsizing implies that the anticipation-phase is a stressful time in the downsizing process for the exposed employees. The results however, suggested an even more prominent increase psychotropic drug purchases across the period of downsizing and the following years among those who continued to work in the same workplace, i.e., the stayers. This also implies that an organizational downsizing can have long-term consequences on mental health for those who stay in the organization, which did not appear to be attributed to health selection, indicating a need for preventive measures.” 

Altogether in the discussion, conclusions and abstract I sometimes found it difficult to say what your main conclusions for the leavers and the unemployed were. Perhaps these two groups could be included the conclusions, as well as add more detail on the main findings in the beginning of the discussion or the abstract?

See our suggestion above, where we’ve tried to address this issue by more explicitly include how unemployed and changers were affected in the overall conclusion. We also revised the abstract slightly to be more explicit about the changers and the unemployed, i.e., the leavers. See the abstract and below:

“This study indicates that being exposed to downsizing is associated with increased use of sedatives and anxiolytics, before the event among those who leave, but especially thereafter for employees who stay in the organization.”

We further added the following sentence in the discussion section concerning the most important findings for these two groups, which is placed as the second paragraph in the discussion section, p. 20

“During the period prior to the downsizing event, often referred to as the anticipation phase, it was primarily the unemployed and the changer groups that seemed to experience an increase in psychotropic drug purchases. The findings are in accordance with previous studies showing that the anticipation phase before a job change or unemployment has predominantly detrimental health effects [29-31].”

7. PLOS authors have the option to publish the peer review history of their article (what does this mean?). If published, this will include your full peer review and any attached files.

Do you want your identity to be public for this peer review? For information about this choice, including consent withdrawal, please see our Privacy Policy.

Reviewer #2: No

Reviewer #3: No

---

## [Decision Letter · Decision Letter 2]

22 Nov 2023

Downsizing and purchases of psychotropic drugs: a longitudinal study of stayers, changers and unemployed

PONE-D-23-00866R2

Dear Dr. Blomqvist,

We’re pleased to inform you that your manuscript has been judged scientifically suitable for publication and will be formally accepted for publication once it meets all outstanding technical requirements.

Kind regards,

Ali B. Mahmoud, Ph.D.

Academic Editor

PLOS ONE

Additional Editor Comments (optional):

Reviewers' comments:

Reviewer's Responses to Questions

**Comments to the Author**

1. If the authors have adequately addressed your comments raised in a previous round of review and you feel that this manuscript is now acceptable for publication, you may indicate that here to bypass the “Comments to the Author” section, enter your conflict of interest statement in the “Confidential to Editor” section, and submit your "Accept" recommendation.

Reviewer #3: All comments have been addressed

2. Is the manuscript technically sound, and do the data support the conclusions?

Reviewer #3: Yes

3. Has the statistical analysis been performed appropriately and rigorously? 

Reviewer #3: Yes

4. Have the authors made all data underlying the findings in their manuscript fully available?

Reviewer #3: Yes

5. Is the manuscript presented in an intelligible fashion and written in standard English?

Reviewer #3: Yes

6. Review Comments to the Author

Reviewer #3: I have no further comments. The authors make a strong case in arguing for publishing the revised version without making more major changes.

7. PLOS authors have the option to publish the peer review history of their article (what does this mean?). If published, this will include your full peer review and any attached files.

Reviewer #3: No

---

## [Editor Report · Acceptance letter]

28 Nov 2023

PONE-D-23-00866R2 

Downsizing and purchases of psychotropic drugs: a longitudinal study of stayers, changers and unemployed 

Dear Dr. Blomqvist:

I'm pleased to inform you that your manuscript has been deemed suitable for publication in PLOS ONE. Congratulations! Your manuscript is now with our production department. 

Kind regards, 

on behalf of

Dr. Ali B. Mahmoud 

Academic Editor

PLOS ONE